# Predictors of Early Thrombotic Events in Adult Patients with Acute Myeloid Leukemia: A Real-World Experience

**DOI:** 10.3390/cancers14225640

**Published:** 2022-11-17

**Authors:** Giovangiacinto Paterno, Raffaele Palmieri, Vittorio Forte, Valentina Del Prete, Carmelo Gurnari, Luca Guarnera, Flavia Mallegni, Maria Rosaria Pascale, Elisa Buzzatti, Valeria Mezzanotte, Ilaria Cerroni, Arianna Savi, Francesco Buccisano, Luca Maurillo, Adriano Venditti, Maria Ilaria Del Principe

**Affiliations:** 1Hematology, Department of Biomedicine and Prevention, University Tor Vergata, 00133 Rome, Italy; 2Department of Clinical Sciences and Translational Medicine, University of Rome Tor Vergata, 00133 Rome, Italy

**Keywords:** acute myeloid leukemia, thrombosis, venous thromboembolism, thrombosis risk score, survival

## Abstract

**Simple Summary:**

The reported incidence of thrombotic events (TE) in non-promyelocytic acute myeloid leukemia (AML) patients varies in the literature from 2% to 13%. The aim of our retrospective study was to assess the incidence of TE in a real-word population of AML patients to determine the impact of TE on survival and to recognize risk factors for early venous thromboembolism (VTE). We observed a TE incidence of 14.6% among 300 patients with newly diagnosed AML. Arterial TE but not VTE was associated with a poorer OS. Furthermore, we observed a higher relapse rate among patients experiencing a VTE. We recognized platelets count >50 × 10^9^/L, presence of comorbidities and a previous history of TE as risk factors for early VTE development. Accordingly, we proposed a score combining these factors that may help in implementing strategies to manage patients at higher risk of early thrombotic complications.

**Abstract:**

Information regarding the incidence and the prognostic impact of thrombotic events (TE) in non-promyelocytic acute myeloid leukemia (AML) is sparse. Although several risk factors associated with an increased risk of TE development have been recognized, we still lack universally approved guidelines for identification and management of these complications. We retrospectively analyzed 300 consecutive patients with newly diagnosed AML. Reporting the incidence of venous TE (VTE) and arterial TE (ATE) was the primary endpoint. Secondarily, we evaluated baseline patient- and disease-related characteristics with a possible influence of VTE-occurrence probability. Finally, we evaluated the impact of TE on survival. Overall, the VTE incidence was 12.3% and ATE incidence was 2.3%. We identified three independent predictors associated with early-VTE: comorbidities (*p* = 0.006), platelets count >50 × 10^9^/L (*p* = 0.006), and a previous history of VTE (*p* = 0.003). Assigning 1 point to each variable, we observed an overall cumulative incidence of VTE of 18.4% in the high-risk group (≥2 points) versus 6.4% in the low-risk group (0–1 point), log-rank = 0.002. Overall, ATE, but not VTE, was associated with poor prognosis (*p* < 0.001). In conclusion, TE incidence in AML patients is not negligible. We proposed an early-VTE risk score that could be useful for a proper management of VTE prophylaxis.

## 1. Introduction

The association between cancer and thrombotic events (TE) has been well-known for more than a century, as it was firstly reported by Jean Baptiste Bouillaud in 1823 [1]. Since then, several studies have described this complication of patients with solid tumors, who are estimated to have a seven-fold increased risk of developing TE as compared to the general population [2,3]. In these patients, TE correlate with a worse prognosis in terms of both overall (OS) and relapse-free survival (RFS) [4,5], representing the second leading cause of death [6].

In patients with hematological malignancies (HM), treatment-related morbidity and mortality are presumed to be almost exclusively related to infection and bleeding, given the common condition of thrombocytopenia and neutropenia at diagnosis and during the course of the disease. Conversely, in such a setting TE has been considered rare and has not received a great deal of attention. However, recent studies indicate a non-negligible risk of thrombosis in HM patients, similar or even higher than in subjects with solid cancers [7,8]. The incidence of TE is well known in some HM such as myeloma (up to 25% without thromboprophylaxis and during therapy with thalidomide) [9], non-Hodgkin lymphoma (6.5%), and Hodgkin disease (4.7%) [10]. Furthermore, the frequency of arterial thrombotic events (ATE) and venous thromboses (VTE) is approximately 11–25% in essential thrombocythemia, 12–39% in polycythemia vera and 10% in acute promyelocytic leukemia (APL) [11,12,13].

Available information about thrombosis in non-promyelocytic acute myeloid leukemia (AML) is scarce, with a reported incidence of TE ranging from 2% to 13% [14,15,16,17]. In such a scenario, the pathogenesis of TE is multifactorial with leukemia cells exerting a prothrombotic effect through the release of fibrinolytic and proteolytic factors, inflammatory cytokines and over-expression of tissue factor on the cell surface of leukemic cells [18,19,20]. Active therapy represents a further predisposing element. Indeed, chemotherapy-induced cell lysis results in a massive release of these procoagulants which, together with the hemodynamic changes and CVC-related vessel injuries, may explain the reported increased rate of TE also in AML patients, in whom the common occurrence of thrombocytopenia may lead to underestimation of this complication [21]. Moreover, patients with AML often present with a hypercoagulable state or chronic disseminated intravascular coagulation (DIC), even in the absence of active thrombosis [22,23]. Therefore, an accurate estimate of an individual patient’s TE risk is essential to decide whether anticoagulation is needed, and to balance the benefit of thromboprophylaxis against the hemorrhagic risk due to the underlying disease- and chemotherapy-induced thrombocytopenia.

The primary aim of our study is: (1) to assess the incidence of TE in a real-word population of non-promyelocytic AML patients; (2) to assess the association between baseline disease and patients’ characteristics and TE; (3) to generate a clinical score predicting the risk of VTE; (4) to determine the impact of TE on survival.

## 2. Materials and Methods

### 2.1. Patients’ Characteristics

We reviewed the records of 300 consecutive newly diagnosed adult patients (≥18 years of age) with non-promyelocytic AML, admitted between January 2010 and December 2020 to the Hematology Department of our institution. Baseline characteristics included in the analysis were date of AML diagnosis, age, gender, body mass-index (BMI), genetics/cytogenetics, blood count and serum chemistry. Information about chemo-therapeutic regimens, date of complete remission (CR)/resistance, date of relapse, date of death or last follow-up, comorbidities, date of TE diagnosis and treatment and TE characteristics were also collected. Comorbidities were defined as the simultaneous presence of one or more diseases or medical conditions, not related to the underlying AML. Routine blood tests, including hemoglobin, white blood cell (WBC) and platelet count (PLTc), were carried out on ethylenediaminetetra-acetic acid (EDTA)-anticoagulated blood samples. Assessment of the coagulation parameters, such as prothrombin time (PT), activated partial thromboplastin time (APTT), fibrinogen (Clauss method) and D-dimer (immuno-turbidimetric method) coagulation tests were performed on citrated plasma tubes. The presence of DIC was assessed according to the 2018 revision of the International Society of Thrombosis and Haemostasis (ISTH) scoring system for DIC [24] as follows: platelet count (≥100 × 10^9^/L = 0; 50–99.999 × 10^9^/L = 1; <50 × 10^9^/L = 2), fibrinogen level (≥100 mg/dL = 0; <100 mg/dL = 1), PT prolongation above upper limit of normal (ULN) (<3 s = 0, 3–6 s = 1, >6 s = 2), and D-dimer (<3000 ng/mL = 0, 3000–7000 ng/mL = 2, >7000 ng/mL = 3). A score sum ≥4 denotes the presence of an overt DIC. Patients were stratified according to the ELN 2017 genetic/cytogenetic risk assessment [25]. Depending on age, performance status or availability of clinical trials for the treatment of AML, patients received intensive chemotherapy (mostly “7 + 3”-like schedule or fludarabine based regimens), non-intensive chemotherapy (such as low dose of cytarabine or hypomethylating agents) or supportive therapy [26]. The study was approved by the Internal Review Board of our Institution and conducted in accordance with the regulations set forth by the Declaration of Helsinki.

### 2.2. Thrombotic Events Diagnosis and Management

TE were defined as a composite of first VTE (not including superficial vein thrombosis) or ATE at any site. VTE were diagnosed by Doppler ultrasonography, computed tomography, or magnetic resonance imaging. Myocardial infarction (MI) was diagnosed according to clinical, enzymatic and electrocardiographic criteria as per well established guidelines [27]. The diagnosis of ATE other than MI was made by computed tomography. Patients with VTE and PLTc ≥30 × 10^9^/L were treated with low-molecular-weight heparin (LMWH) for 3 to 6 months [28,29]. The dose of LMWH was reduced by 50% if PLTc was between 30 and 50 × 10^9^/L and discontinued or not initiated when PLTc was <30 × 10^9^/L. Patients with a previous history of MI were prophylactically treated with antiplatelet drugs if PLTc >50 × 10^9^/L [30].

### 2.3. Statistical Analyses

Comparisons between groups were performed to assess differences in biological and clinical data using the Chi-squared test or Fisher’s exact test for categorical data and Mann-Whitney and Kruskal-Wallis test in case of continuous variables. Risk factors associated with the development of VTE or ATE were analyzed by logistic regression models. Patients who presented with a TE before chemotherapy initiation were considered to belong to the group receiving supportive therapy. We dichotomized potentially significant continuous variables at an optimal cut-off point identified with ROC analysis and the Youden index. Hosmer and Lemeshow’s stepwise strategies were applied for model building: potential independent variables with *p*-value < 0.25 were included in the initial full model. All tests were 2-sided, accepting *p* < 0.05 as a statistically significant value. Statistically significant variables (*p* < 0.05) were included in the final model. The model was evaluated using Hosmer–Lemeshow tests and pseudo-R2 measures. The final prediction score was derived based on weighed variables in the final model and compared using Kaplan–Meier survival analysis with log- rank tests to assess differences in identified categories. Internal validation was conducted using nonparametric bootstrapping methods, obtaining estimates of the variability of the β-coefficients for the parameters in the regression equations. To evaluate the predictive accuracy of the model, a Harrell’s statistics was performed and a C-index was calculated. OS was defined as the time from the diagnosis to death from any cause, and in patients with TE as the time from TE diagnosis to death from any cause. RFS was defined as the time from the achievement of CR after induction therapy to disease-relapse or death from any cause, whichever came first. Probabilities of OS and RFS were calculated using the Kaplan-Meier method. All statistical analyses were performed using SPSS v28.

## 3. Results

Overall, the median age at AML diagnosis was 64 years (range 21–90), with a slight male predominance (61%, *n* = 178). Among the 300 patients, 69% had de novo AML, 19% had secondary AML, and 12% had therapy-related AML. According to the ELN 2017 risk stratification, 17.3% of the cases were classified as favorable, 41.4% intermediate and 28.3% adverse risk, whereas 13% were not classifiable. At least one comorbidity was present in 67.7% of the patients, including hypertension (41%), autoimmune disorders (15%) and diabetes (10.7%). Two or more cardiovascular comorbidities were observed in 24.7% of the patients with 6.7% reporting previous coronary diseases or myocardial infarction and, therefore, receiving antiplatelets medication. An additional 4.3% of patients were receiving anticoagulation for other indications (atrial fibrillation, chronic VTE, cerebrovascular ischemia, or heart failure). At baseline, 20% of cases showed an ISTH-DIC score ≥4. As to treatments, 58.3% received intensive chemotherapy, 19.7% non-intensive strategies and 22% only supportive care. The demographic and disease characteristics of the study cohort are summarized in Table 1.

During the overall study period (median follow-up 30 months, range 2–126 months), a total of 44/300 patients (14.6%) experienced a TE, the majority of which (12.3%) were VTE, with 2.3% being ATE. We observed 28 CVC-related thromboses (CRT), 5 PE (2 occurring in the context of deep vein thrombosis of the lower extremities and 1 of the arm), 2 deep vein thromboses of the lower extremities without pulmonary embolism, 2 hepatic veins thrombosis, 6 MI, and 1 peripheral arterial thrombosis. Patients suffering from TE had a median age at AML diagnosis of 71.5 years (range 30–82). None of the 44 patients had received anticoagulant prophylaxis in the two weeks preceding TE occurrence. Overall, the cumulative incidence of VTE at 45 days, 3 months, 6 months, and 2 years from AML presentation was 9.2%, 11.1%, 13.8% and 15%, respectively, while the cumulative incidence of ATE was 2.8%, 3.3%, and 4.8%. The majority of TE (82%, 30 VTE and 6 ATE) occurred within 45 days from AML presentation (median 25 days, range 1–40 days), 58% of which (19 VTE and 2 ATE) before starting AML-therapy. The characteristics of TE are detailed in Table 2. Eight patients developed TE after 45 days from AML presentation: 3 VTE and 1 ATE at AML relapse, 2 VTE after of HSCT and 2 VTE under hypomethylating agents. These patients were excluded from the analysis because of potential bias, related to a myriad of factors treatment- and not-treatment related.

The presence of one or more comorbidities (OR: 4.729; 95% CI 1.398–16.000; *p* = 0.006), a previous history of VTE (OR: 7.514; 95% CI 2.221–25.420; *p* = 0.003) and a PLTc at diagnosis higher than 50 × 10^9^/L (OR: 3.287; 95% CI 1.413–7.643; *p* = 0.006) were associated with an increased risk of VTE, within 45 days from AML diagnosis. Age, sex, BMI, leucocyte count, hemoglobin level, an ISTH-DIC score ≥4, ELN risk stratification and cytogenetic and genetic abnormalities were not associated with an increased risk of VTE development. No difference in VTE rate was observed in patients with or without CVC (*n* = 25/252, 10% vs. 5/48, 10% respectively, *p* = 1). Once resolved, no recurrence of VTE was recorded in any of the patients. With the limitation of the small sample size, ATE were associated with the presence of two or more cardiovascular comorbidities (i.e., hypertension, previous IM or ischemic stroke, peripheral arterial disease, atrial fibrillation etc.), (*p* < 0.001). Despite the cardiovascular comorbidities, it must be noted that 3 patients had normal cardiac function (of which, 2 were given intensive chemotherapy, 1 non-intensive), 1 died soon thereafter AML diagnosis (thereby cardiac function was not evaluated), 2 had existing cardiac impairment. Excluding patients developing TE prior to AML therapy initiation, no difference in TE rates was seen in patients receiving intensive chemotherapy (12/161, 7.5%) vs. non-intensive therapy (3/54, 5.6%; OR 0.730, 95% CI 0.198–2.69; *p* = 0.869).

### 3.1. Development of Venous Thromboembolism Prediction Score

In the attempt to identify patients at higher risk of TE, amenable to early interventions, we built a practice prediction model using 22 potential baseline predictors. Among the risk factors considered, only 3 were included in the final model. These were a previous VTE history, presence of comorbidities, and PLTc > 50 × 10^9^/L at the time of AML diagnosis. Non- parametric bootstrapping confirmed the significance of the selected variables (Table 3). A final prediction score was based on weighed variables assigning 1 point for each of the aforementioned variables. A score summing 2 or more defined patient at high-risk of VTE. When applying this score to our cohort, 34.7% were at high-risk for VTE. This resulted in an overall cumulative incidence of VTE within 45 days from AML presentation of 18.4% in the high-risk and 6.41% in the low-risk group (Figure 1). The Kaplan-Meier survival analysis (log-rank *p* = 0.002) and the C-index for the model (0.641; 95% CI: 0.534–0.747) demonstrated a significant discrimination between the two categories and an efficient predictive accuracy.

### 3.2. Impact of Thrombosis on Survival

We finally explored whether the occurrence of early TE would affect survival outcomes in patients with AML. With a median follow-up of 30 months, the overall OS rate was 28.4% at 2 years. No differences were observed in terms of 2-year OS (28.7% vs. 31.3%, HR 0.88 *p* = 0.552) rates between patients experiencing or not VTE. We did not record any VTE-related death in our series of patients. In contrast, five of six patients experiencing ATE died within 12 days from the event (HR 9.23, *p* < 0.001). The OS curves are shown in Figure 2. Finally, while no differences in terms of CR rates were observed between patients developing VTE or not (63% vs. 48% *p* = ns), we noticed that patients developing VTE had a shorter RFS (Figure 3) when compared to the non-VTE counterpart (2-years RFS 31% vs. 44%, *p* = 0.046). Timing of TE (prior to vs. during treatment) did not affect patients’ outcome (HR 1.26 *p* = 0.539).

## 4. Discussion

The pathogenesis of thrombosis in patients with cancer is complex and involves both patient and disease-related factors. Parameters known to be associated with an increased risk of TE among AML patients include comorbidities, age, sex, use of hematopoietic growth-factors, presence of CVC, PLTc and prior history of venous thromboembolism [31]. To date, literature evidence concerning the incidence of TE in AML is limited, mostly based on small series of patients, non-homogeneously evaluated from a diagnostic point of view for TE, such as the inclusion of only symptomatic VTE, exclusion of CRT, or heterogeneity of studies population [14,15,16,32,33]. Furthermore, the frequent condition of thrombocytopenia contraindicates an adequate delivery of anticoagulant therapy and further contributes to lack of clarity when precisely assessing the thrombotic risk in such patients.

We evaluated the rate and characteristics of TE in a large cohort of patients with AML. Moreover, we included patients undergoing either intensive or less intensive treatments in order to offer a comprehensive overview of TE incidence in a real-world scenario. In line with data reported on patients with other cancer types, we observed a global rate of TE incidence of 15%, and a risk of VTE development during the early phase of the disease of 10% [7,14,15]. Having one or more comorbidities, a prior history of VTE and a PLTc higher than 50 × 10^9^/L at AML presentation were associated with a greater risk of early VTE. Following the evidence suggested by Al-Ani et al. [34], we built a simple tool to assess the VTE risk in AML patients that categorizes patients into two discrete categories. Based on this model, we established that the overall cumulative incidence of VTE was 18.4% in the high-risk group (≥2 points) and 6.4% in the low-risk group (0–1 points).

Indeed, prior history of VTE is a well-known major risk factor for developing VTE in cancer patients [5,35]. As reported by Vu et al. [32], 7% of AML patients who developed VTE had a prior history of VTE, vs. only 2.1% in cases without previous VTE. Several studies have also identified an association between burden of comorbidities and increased risk of cancer-associated thromboses [36,37], even in AML patients [14].

PLTc has been shown to have significant association with cancer and thrombosis. The Khorana score [38], commonly used to stratify the risk of VTE in patients with cancer undergoing chemotherapy, indicates that a PLTc of 350 × 10^9^/L is a risk factor for VTE [33]. Such a cut-off is not optimal to quantify the risk of thrombosis in AML patients since the very frequent baseline condition of thrombocytopenia precludes its applicability. Studies specifically focused on patients with acute leukemias [17,34] found that a baseline PLTc > 50 × 10^9^/L was an independent risk factor for VTE development. Given the propensity of AML to trigger coagulopathy, Libourel et al. [16] evaluated the role of the ISTH-DIC score [39] to predict the risk of thrombosis in AML patients. The ISTH-DIC score was developed as a diagnostic algorithm for DIC for patients with a clinical suspicion of coagulative disorders (e.g., excessive bleeding in the setting of malignancy, severe infection or sepsis, obstetric complications, trauma). However, this score takes into account PLTc, fibrinogen levels, prothrombin time, and D-Dimer. Therefore, its translation into the AML scenario is someway troublesome as low PLTc (<50 × 10^9^/L) is a very common finding and typically related to AML itself [40]. Moreover, delivery of chemotherapy contributes to alter the evaluation of the severity of DIC due to leukemic cells lysis and tissue cellular damage. For these reasons, inherently linked to the nature of the disease and to its treatment, the ISTH-DIC score fails to reliably predict risk of VTE development in AML. While Libourel showed that the occurrence of TE was significantly higher in patients with laboratory evidence of DIC (DIC score ≥ 5) prior to the initiation of treatment, other studies confirmed the limitation of ISTH-DIC score in VTE-prediction in AML patients [17]. Our study did not show a significant association between VTE and ISTH DIC score. Finally, we confirmed the absence of correlation between the presence of cytogenetic and genetic abnormalities and the development of VTE.

Despite the high number of CRT, we did not observe a statistically significant cor-relation between the presence of CVC and the incidence of VTE. The incidence of CRT observed in our cohort is comparable to that observed in the literature [41]. CRT occurring during the first few days after device insertion are to be related to the local venous injury, reduction of blood flow and deposition of fibrin on the catheter surface. On the other hand, late onset CRT may be related to blood stream infections or to disease-related coagulopathy [42].

In patients with solid tumors, TE are associated with adverse short-term and long-term prognosis [4,43]. In patients with acute leukemia, only a retrospective, registry-based study by Poh et al. [44] evaluated the effect of upper-extremity TE on survival. Analyzing 5072 patients (3252 AML and 1820 ALL) treated with intensive chemotherapy and registered in the California Cancer Registry, the authors found that upper-extremity TE was an independent predictor for increased leukemia-associated and overall mortality in both AML and ALL. The authors speculated that the higher mortality of patients with TE was a surrogate of disease severity and not only the ultimate cause of death. In our study, we did not observe VTE-related death, there was no difference in CR rate between patients with or without VTE, and VTE was not associated with a poorer OS. At the same time, we observed a higher relapse rate among patients experiencing a VTE. In agreement with Poh et al., we could speculate that ill-defined biologic factors may still contribute to thrombogenesis and that VTE could reflect a more aggressive, relapse-prone disease and serve as an additional relapse risk factor. This is different for patients suffering from ATE: 6 patients experienced a MI in the first 45 days after the diagnosis of AML, 5 of whom died within 12 days from the event. These data are consistent with the current literature [45], and could be explained by the different pathogenesis underlying ATE compared to VTE.

We acknowledge the limitations of our study such as its retrospective nature and the absence of an external validation of the proposed scoring system, potentially limiting the broad applicability of the score prior to independent validation studies. However, we believe that the homogeneity of our series, its consecutive recruitment, the management based on standardized diagnostic and therapeutic procedures and the long-term follow up, reflecting 10 years of clinical practice, constitute a valuable strength of our work.

## 5. Conclusions

Our study highlights that a non-negligible proportion of patients with AML can develop TE, especially of venous origin. Early identification of patients at higher risk of such a complication is challenging and represents an unmet need with potentially relevant clinical consequences. In fact, a possible great risk of TE is made ever more complicated by the AML intrinsic proclivity to bleeding, which precludes unconditionally the use of primary thromboprophylaxis. Therefore, identification of predictors of TE are mandatory for adequate clinical management. As the thrombotic risk-scoring systems used for non-HM does not apply to AML patients, we identified three independent predictors to quantify patients’ VTE risk. This score may help in implementing strategies to manage patients at higher risk of early thrombotic complications. Once could speculate as a potential approach to use higher platelet transfusion thresholds (to maintain a PLTc > 30 × 10^9^/L) along with prophylactic doses of LMWH and/or nonpharmacologic prophylaxis in selected cases.

Further investigation and an external validation of our prediction score on larger cohorts of patients are warranted. Finally, prospective studies investigating the pathogenesis underpinning the occurrence of VTE in AML, its correlations with prognosis, and the proper management of prophylaxis and therapeutic anticoagulation in this setting of patients may shed light onto such oftentimes-neglected AML complication.

## Figures and Tables

**Figure 1 cancers-14-05640-f001:**
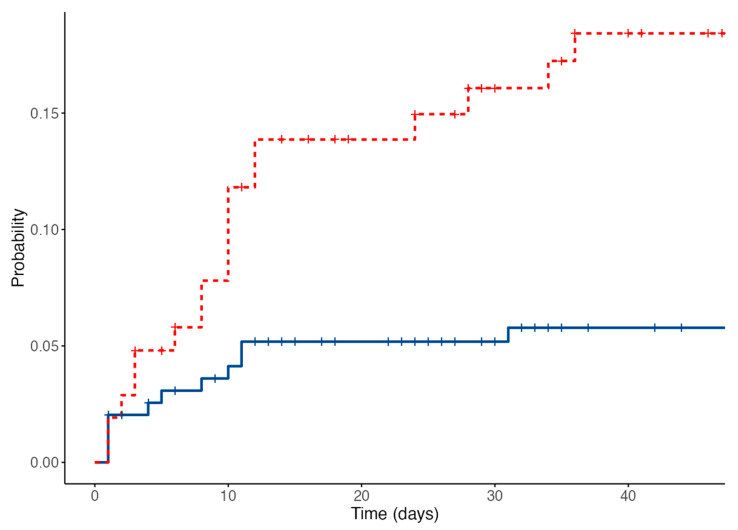
Overall cumulative incidence of venous thromboembolic events, low risk patients in blue, high risk patients in red.

**Figure 2 cancers-14-05640-f002:**
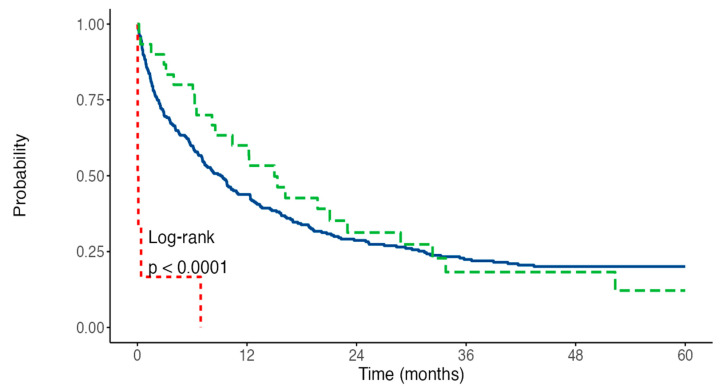
Overall Survival, patients with no TE in blue, VTE in green, ATE in red.

**Figure 3 cancers-14-05640-f003:**
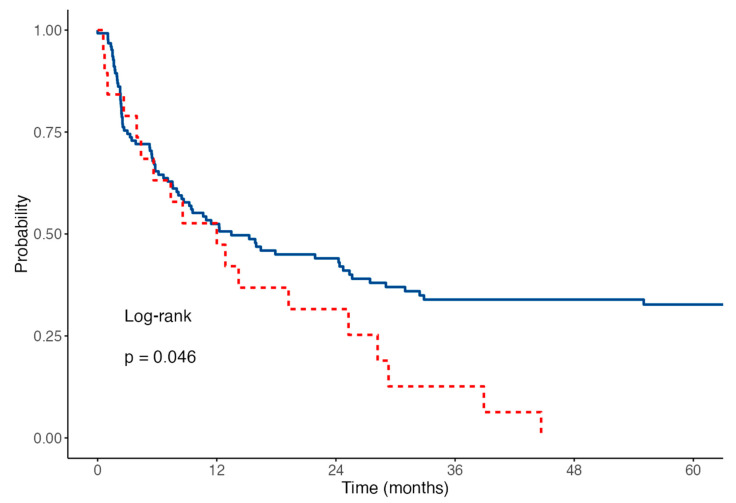
Relapse-free survival, patients with VTE in red, no VTE in blue.

**Table 1 cancers-14-05640-t001:** Baseline patient’s characteristics.

	No TE *n* = 264	TE *n* = 36	*p*	VTE, *n* = 30	*p*	ATE, *n* = 6	*p*
Sex M/F	159/105	24/12	0.585	18/12	1	6/0	0.085
Median Age (interval)	64 (21–90)	64 (30–82)	0.998	64 (30–82)	0.708	66 (53–77)	0.584
**AML Classification**							
de Novo AML	183	24	0.203	20	0.137	4	0.59
Treatment-related AML	35	2	0.219	1	0.195	1	0.318
Secondary-AML	46	10	0.152	9	0.179	1	0.876
**ELN 2017-risk stratification**							
Favorable	46	6	0.908	5	0.919	1	0.93
Intermediate	108	16	0.803	13	0.862	2	0.885
Adverse	73	12	0.665	10	0.696	2	0.868
Non-Classifiable	37	2		2		1	
FLT3	49	8	0.652	7	0.621	1	1
NPM1	54	5	0.379	4	0.467	1	1
Comorbidities ≥ 1	171	32	0.004	27	0.006	6	0.182
Cardiovascular comorbidities ≥ 2	58	16	0.006	10	0.266	6	<0.001
BMI ≥ 1	24	5	0.367	4	0.511	1	0.46
ISTH-DIC score ≥ 4	56	5	0.382	5	0.811	0	0.605
Smokers	107	18	0.286	15	0.337	3	0.696
History of VTE	7	5	0.008	5	0.003	0	
Central Venous Catheter	221	31	0.813	25	1	6	
Hemoglobin Count g/dL	8.55 (3.5–13.8)	8.95 (5.2–13.9)	0.371	8.7 (5.2–11.7)	0.939	9.7 (8.4–13.9)	0.059
Leukocyte Count × 10^9^/L	9.11 (10–472.3)	8.77 (0.47–200)	0.614	7.83 (0.47–200)	0.708	6.93 (0.82–26)	0.286
Platelet Count × 10^9^/L	44.5 (3–1130)	71 (5–597)	0.032	76 (5–597)	0.05	44.5 (29–95)	0.998
Serum LDH level, U/L	350 (70–4000)	300 (120–1687)	0.425	292 (120–1687)	0.262	364 (234–811)	0.707
Serum Fibrinogen level mg/dL	364 (63–2878)	373 (82–1025)	0.842	364 (82–1025)	0.677	506 (332–872)	0.027
ATIII, %	92 (25–142)	97.3 (70–125)	0.147	97.3 (71–125)	0.2	88.3 (70–115)	0.727
PT, seconds	14.6 (11.1–29.6)	14.5 (11.4–20.4)	0.439	14.85 (11.4–20.4)	0.66	14.7 (12.4–16.5)	0.981
aPTT, ratio	0.980 (0.59–1.86)	0.925 (0.68–2.01)	0.136	0.935 (0.68–2.01)	0.379	0.950 (0.8–0.94)	0.076
Serum D-Dimer level ng/mL	834 (52–121,450)	886 (77–48,065)	0.936	999 (77–48,065)	0.948	1279 (532–20,650)	0.324

All data are the median value (range) unless otherwise indicated. AML, acute myeloid leukemia; ELN. European Leukemia Net; BMI, body mass index; ISTH-DIC score, International Society of Thrombosis and Haemostasis- disseminated intravascular coagulation score; VTE, venous thromboembolism; ATE, arterial thrombosis; LDH, Lactate Dehydrogenase; ATIII, Antithrombin; PT, Prothrombin Time; aPTT, activated Partial Thromboplastin Time. Statistically.

**Table 2 cancers-14-05640-t002:** Characteristics of thrombotic events occurred within 45 days from acute myeloid leukemia diagnosis.

	Age	Sex	Type of TE	Onset from AML Diagnosis (Days)	History of VTE	n° of Comorbidities	TE Pre/Post AML-Therapy
1	66	M	hepatic veins thrombosis	1	no	2	pre
2	47	F	CRT	8	no	2	pre
3	63	M	CRT	10	no	1	post
4	68	M	DVT+PE	1	no	3	pre
5	53	M	MI	1	no	2	pre
6	44	F	CRT	11	no	1	post
7	71	F	PE	12	no	2	post
8	63	M	MI	35	no	2	post
9	81	M	CRT	18	no	2	pre
10	70	M	DVT	1	no	2	pre
11	68	M	CRT	4	no	2	pre
12	49	F	CRT	11	no	2	post
13	64	F	CRT	10	no	1	pre
14	30	M	CRT	8	no	0	pre
15	58	M	CRT	5	no	2	pre
16	63	F	DVT	1	yes	1	pre
17	64	M	DVT	3	no	3	pre
18	76	M	CRT	8	no	3	post
19	61	M	CRT	6	no	1	post
20	73	F	PE	40	no	1	post
21	59	M	CRT	36	yes	1	post
22	40	F	DVT+PE	10	no	0	post
23	82	F	CRT	28	no	2	pre
24	41	M	CRT	34	no	0	post
25	55	M	CRT	12	no	1	pre
26	64	M	MI	24	no	2	post
27	75	M	CRT	31	yes	1	post
28	72	M	MI	13	no	1	post
29	69	F	CRT	2	no	1	pre
30	73	M	CRT	6	yes	1	pre
31	48	F	hepatic veins thrombosis	1	no	0	pre
32	72	M	CRT	3	no	1	pre
33	55	M	CRT	10	no	3	pre
34	68	M	MI	24	no	2	pre
35	67	F	CRT	10	yes	3	pre
36	77	M	MI	25	no	4	post

TE, thrombotic event; AML, acute myeloid leukemia; VTE, venous thrombotic event; M, male; F, female; CRT, catheter-related thrombosis; MI, myocardial infarction; DVT deep-venous thrombosis; PE, pulmonary embolism.

**Table 3 cancers-14-05640-t003:** Logistic regression model for occurrence of venous thromboembolism within 45 days from acute myeloid leukemia diagnosis.

Risk Factor	β-Coefficient	95% Coefficient Interval for β-Coefficient	*p*-Value	*p*-Value (Bootstrapped)
Platelet count > 50 × 10^9^/L at baseline	3.238	1.361–7.704	0.008	0.004
History of venous thromboembolism	5.447	1.521–19.514	0.009	0.004
Presence of one or more comorbidities	3.974	1.150–13.734	0.029	0.021

## Data Availability

The data presented in this study are available on request from the corresponding author.

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
