# Peer review of "Predictors of Early Thrombotic Events in Adult Patients with Acute Myeloid Leukemia: A Real-World Experience"

_cancers, 2022, doi:10.3390/cancers14225640_

Round 1

Reviewer 1 Report

In this study, authors evaluated retrospectively the VTE and ATE in patients with AML. The incidence of thrombotic events in AML has been already reported, so the novelty of this study is mainly based on the identification of a simple tool to predict risk of TE and ATE in AML based on comorbidities, PLT >50 and previous history of VTE. The authors also describe no impact in OS but in DFS in patient with high risk vs low risk according to their own scoring system. 

The study is interesting but I have few questions:

1. was the incidence higher in patient with CVC comparing with non CVC, considering the fact that not all the patient population was on intensive chemotherapy?

2. Was the risk of TE higher in granulated blasts  AML? 

3. Was the ECHO function normal in all patient with MI, can the authors excluded a predisposing cardiomyopathy? 

4. All patients experiencing MI had antracyclines ?

5. was the risk higher in patient treated with intensive chemotherapy vs non intensive?

6. how many patients did have TE prior to commencing treatment ? Was the outcome different between patient with TE prior to treatment and during treatment ?

Author Response

  1. Was the incidence higher in patient with CVC comparing with non CVC, considering the fact that not all the patient population was on intensive chemotherapy?

We added the following sentence at line n.192-193 “…No difference in VTE rate was observed in patients with or without CVC (n=25/252, 10% vs 5/48, 10% respectively, p=1)....“

  1. Was the risk of TE higher in granulated blasts AML? 

Unfortunately, this information was not available for all patients and particularly was only present for very few of the 44 patients experiencing TEs. Therefore, the little numbers precluded any further statistical test. We thank the reviewer for giving us a suggestion for our future studies.

  1. Was the ECHO function normal in all patients with MI, can the authors excluded a predisposing cardiomyopathy? 

We added the following sentence at line 196-199 “…Despite the cardiovascular comorbidities, it must be noted that 3 patients had normal cardiac function (of which, 2 were given intensive chemotherapy, 1 non-intensive), 1 died soon thereafter AML diagnosis (thereby cardiac function was not evaluated), 2 had existing cardiac impairment…”.

  1. All patients experiencing MI had antracyclines?

Two patients died because of MI before treatment initiation, 2 patients received anthracyclines-based induction therapy and 2 non-intensive treatment.

  1. was the risk higher in patient treated with intensive chemotherapy vs non intensive?

We added the following sentence at line 200-202 “…Excluding patients developing TE prior to AML therapy initiation, no difference in TE rates was seen in patients receiving intensive chemotherapy (12/161, 7,5%) vs non-intensive therapy (3/54, 5,6%; OR 0,730, 95% CI 0,198-2,69; p=0,869)”

  1. how many patients did have TE prior to commencing treatment? Was the outcome different between patient with TE prior to treatment and during treatment?

 We thank the reviewer for this comment. Overall, 21 patients experienced a TE prior to AML-treatment, while 15 during treatment (line 175-176). We added the following sentences at line n 232-233 “Timing of TE (prior to vs during treatment) did not affect patients’ outcome (HR 1,26 p=0.539)”

Reviewer 2 Report

In this manuscript, the authors provided a retrospective survey on the factors involved in the development of early thrombotic events in AML patients and their implications for patient outcomes. The paper is well-written, the experimental design is appropriate and adequate, and the drawn conclusion is in line with the presented results. The study has certainly some relevant limitations as already cited by the authors.

1.     The main limitation is the lack of a validation cohort since the score is designed and validated on the same series of cases. If no cohort is available to perform validation, I recommend including in the discussion how this may affect the results obtained. 

2.  The precocity of catheter-related thrombosis is striking. There are 12 catheter-related thromboses in the moments prior to leukemia treatment (7 in the first 8 days after diagnosis). Have you found any association (infection, etc) or could the author give any explanation. 

On the other hand, the later catheter-related thromboses occur in older patients who are not given chemotherapy treatment early. A recommendation along these lines could be given. 

3. In Table 1, I recommend checking the expressed leukocyte and platelet counts in ATE (6930...) (44500...) because there must be an error in the units of expression used.

Author Response

  1. The main limitation is the lack of a validation cohort since the score is designed and validated on the same series of cases. If no cohort is available to perform validation, I recommend including in the discussion how this may affect the results obtained. 

We added the following sentence at line 314-316 “…We acknowledge the limitations of our study such as its retrospective nature and the absence of an external validation of the proposed scoring system, potentially limiting the broad applicability of the score prior to independent validation studies…”

  1. The precocity of catheter-related thrombosis is striking. There are 12 catheter-related thromboses in the moments prior to leukemia treatment (7 in the first 8 days after diagnosis). Have you found any association (infection, etc) or could the author give any explanation. On the other hand, the later catheter-related thromboses occur in older patients who are not given chemotherapy treatment early. A recommendation along these lines could be given. 

We added the following sentence at line n 290-296 “…Despite the high number of CRT, we did not observe a statistically significant correlation between the presence of CVC and the incidence of VTE. The incidence of CRT observed in our cohort is comparable to what observed in the literature [43]. CRT occurring during the first few days after device insertion may be related to local venous injuries, reduction of blood flow and deposition of fibrin on the catheter surface. On the other hand, late onset CRT may be related to blood stream infections or to disease-related coagulopathy [44]..”. We also added two references to substantiate the statements in response to the reviewer comments:

  1. Murray, J.; Precious, E.; Alikhan, R.; Catheter-related thrombosis in cancer patients. Br. J. Haematol. 2013, 162, 746-757, doi: 10.1111/bjh.12474; .
  2. Geerts W. Central venous catheter-related thrombosis. Hematology Am Soc Hematol Educ Program 2014, 1: 306-311, doi: 10.1182/asheducation-2014.1.306

3. In Table 1, I recommend checking the expressed leukocyte and platelet counts in ATE (6930...) (44500...) because there must be an error in the units of expression used.

We changed accordingly.

Round 2

Reviewer 1 Report

Thank you and happy with the responses 

Reviewer 2 Report

I agree with the made changes.